# CONTRASTIVE LEARNING VIA EQUIVARIANT REPRESENTATION

## ABSTRACT

Invariant Contrastive Learning (ICL) methods have achieved impressive performance across various domains. However, the absence of latent space representation for distortion (augmentation)-related information in the latent space makes ICL sub-optimal regarding training efficiency and robustness in downstream tasks. Recent studies suggest that introducing equivariance into Contrastive Learning (CL) can improve overall performance. In this paper, we revisit the roles of augmentation strategies and equivariance in improving CL's efficacy. We propose CLeVER (**C**ontrastive **Le**arning **V**ia **E**quivariant **R**epresentation), a novel equivariant contrastive learning framework compatible with augmentation strategies of arbitrary complexity for various mainstream CL backbone models. Experimental results demonstrate that CLeVER effectively extracts and incorporates equivariant information from practical natural images, thereby improving the training efficiency and robustness of baseline models in downstream tasks and achieving state-of-the-art (SOTA) performance. Moreover, we find that leveraging equivariant information extracted by CLeVER simultaneously enhances rotational invariance and sensitivity across experimental tasks, and helps stabilize the framework when handling complex augmentations, particularly for models with small-scale backbones. [1]

## 1 INTRODUCTION

Self-supervised learning (SSL) reveals the relationships between different views or components of the data to produce labels inherent to the data. These labels serve as supervisors for pretext tasks in the pre-training process (Gui et al., 2023). As an unsupervised training strategy, SSL eliminates the reliance on manual labeling, enabling SSL-based methods to achieve superior performance and promising generalization capabilities across many domains (Caron et al., 2021; Devlin et al., 2018; Gui et al., 2023; Oquab et al., 2024).

As a critical methodology in the SSL community, Invariant Contrastive Learning (ICL) generates different views of the same input instance through data augmentation, expecting the backbone model to extract semantic-invariant representations from the different distorted views. However, this semantic-invariance-based approach assumes that only semantics unrelated to the distortions brought by augmentation operations are valuable. In other words, typical ICL methods discard representations affected by augmentation operations. This assumption necessitates careful construction of augmentation strategies to achieve optimal downstream performance (Chen et al., 2020a; Lee et al., 2021; Chen & He, 2021; Chen et al., 2020b; Caron et al., 2021). Moreover, such exquisite augmentation strategies make pre-trained models vulnerable to unseen perturbations (Fig. 1(a)).

As the counterpart to the invariant principle, equivariant-based deep learning is well-studied (Sabour et al., 2017; Batzner et al., 2021; Gerken et al., 2023; Xu et al., 2023; Weiler et al., 2023). Theoretically, a model can learn to be invariant or equivariant as required by the task for which it is trained (Weiler et al., 2023). That is, a model can acquire the invariant or equivariant properties necessary for a task by sufficiently training on a task-specific dataset. However, a naive model needs to explicitly learn these properties, *i.e.*, it needs to be presented with as many possible transformed

---

[1]The anonymized code has been uploaded as supplementary material and will be made publicly available following the double-blind review process.

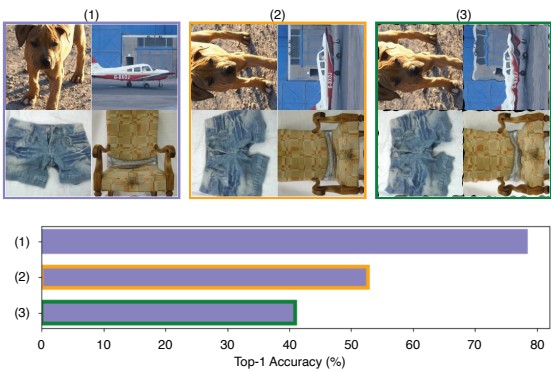
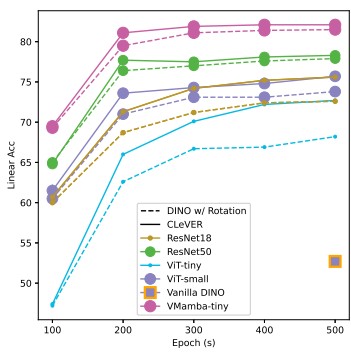

(a) Comparison of DINO (Caron et al., 2021) performance in the face of different perturbations: (1) The samples are in common orientations and states. (2) The samples are in an uncommon orientation or rotated. (3) The samples are in unusual orientations and imaging states.

(b) Comparison of the performance of various backbone models pre-trained with CLeVER and DINO on ImageNet-100. All performances are obtained under rotational perturbation.

Figure 1: CLeVER can introduce a comprehensive robustness improvement for DINO.

states (e.g., poses or positions) to understand the equivariant relationships among these states. This naive approach leads to low training efficiency, high data requirements, weak generalization ability, and non-robustness, which are undesirable. Consequently, many works have emerged that aim to design structurally constrained models by incorporating equivariance (Weiler et al., 2023). Instead of repeatedly learning different views of the same sample, these models automatically generalize their knowledge to all considered transformations. These equivariant-based models typically reduce the number of parameters, complexity, and data requirements while improving training efficiency, prediction performance, generalization, and robustness.

However, prior studies on equivariant-based model design primarily focus on supervised learning and task-specific scenarios. As an unsupervised and general-purpose pre-training strategy, the CL approach cannot realize the introduction of equivariant properties by changing the structural design of the backbone model or the prediction head. Moreover, since the training process of CL deals with pretext tasks, the desired task-specific equivariance in the downstream task remains unknown. Recent works based on Equivariant Contrastive Learning (ECL) introduce rotational equivariance by incorporating rotation into augmentation strategies and integrating temporary modules or architectures into the CL pre-training process (Xiao et al., 2021; Dangovski et al., 2021; Devillers & Lefort, 2022; Bai et al., 2023; Garrido et al., 2023; Gupta et al., 2023; Everett et al., 2024). However, most of these studies assume that equivariance (*e.g.* rotation) is a generic property of downstream tasks, complicating the introduction of more complex equivariances. Notably, a recent study, distortion-disentangled contrastive learning (DDCL) (Wang et al., 2024), proposes an adaptive design that splits output representations and explicitly projects the distortions caused by augmentation into a latent space, thereby leveraging information from augmentations. Since this method does not require distortion-specific modules and architectures, it can be readily extended to more complex augmentation strategies (*e.g.,* rotation and elastic transformation) to introduce more sophisticated equivariances, and has been evaluated on large-scale natural image datasets. However, we observe that the orthogonal loss introduced by DDCL makes the training process unstable and leads to trivial solutions. Therefore, we revisit both the ECL framework and the DDCL method in detail and propose our novel ECL framework, Contrastive Learning Via Equivariant Representation (CLeVER).

In summary, our main contributions are as follows:

- We revisit the ECL framework and DDCL, proposing a simple yet effective regularization loss on the projection head parameters to prevent collapse and trivial solutions when extracting equivariant representations using orthogonal loss.

- We propose a novel ECL framework, CLeVER, based on our regularization loss and an advanced ICL framework DINO (Caron et al., 2021). By adaptively introducing equivariance through augmentation strategies of arbitrary complexity, CLeVER enhances backbone performance, leading to improved training efficiency, generalization, and robustness. CLeVER achieves SOTA results on practical natural images and particularly boosts the performance of models with small to medium-scale backbones.

- Unlike other studies designed for either augmentation invariance or sensitivity, our experiments demonstrate that the equivariant representation (Equivariant Factor) extracted by CLeVER simultaneously enhances both across experimental tasks.

- We employ CLeVER for three mainstream backbone models (ResNet (He et al., 2016), ViT (Dosovitskiy et al., 2020), and VMamba (Liu et al., 2024)) experimentally demonstrating that various types of backbone models can achieve better performance with CLeVER (Fig. 1(b)). Particularly, we find that VMamba-based contrastive learning has outstanding performance on medium-scale data.

## 2 Revisit the Equivariant Representation in CL

Although ICL methods are widely used, the assumption or inductive bias of focusing only on semantic information unrelated to augmentations/distortions raises some potential concerns (Chen et al., 2020a; Chen & He, 2021; Chen et al., 2020b; Caron et al., 2021). To achieve semantic invariant representation in ICL methods, the backbone model needs to learn as many views of the same sample as possible to achieve the multi-view-to-unique-feature mapping. The backbone model needs to memorize as many views as possible for each sample, leading to small-scale models failing to achieve satisfactory performance. In addition, since typical ICL methods disregard all information associated with augmentations/distortions, the backbone model may exhibit poor generalization in downstream tasks that require such semantic information (*e.g.*, color semantics are crucial for classifying food and plants). Furthermore, since ICL methods often carefully select augmentation operations to achieve the best performance scores in common scenarios, pre-trained backbone models lack robustness against unknown perturbations. For example, due to the difficulty of achieving rotational invariance, ICL methods typically avoid choosing rotation as an augmentation operation. As shown in Fig. 1(a), this trade-off results in ICL methods generally struggling to handle rotation as a common perturbation effectively.

Recent studies (Xiao et al., 2021; Dangovski et al., 2021; Devillers & Lefort, 2022; Park et al., 2022; Bai et al., 2023; Garrido et al., 2023; Gupta et al., 2023; Everett et al., 2024) have introduced equivariance into CL methods and proposed several ECL frameworks to address the aforementioned concerns. Although the principles of these works are diverse, these works have several concerns. (a) These frameworks usually focus only on realizing contrastive learning with rotational equivariance, which means they focus on only a sub-problem of ECL, thereby limiting their extensibility and potential. (b) These studies employ some equivariant-specific architectures or pretext task designs, such as adding rotation predictor heads during pre-training to achieve rotation sensitivity (Dangovski et al., 2021; Devillers & Lefort, 2022). Such equivariant-specific designs rely on manual labor and drag down training efficiency. (c) Some existing frameworks are designed to address only a single purpose of either augmentation invariance or sensitivity, and have not yet been extended to practical natural images (Dangovski et al., 2021; Bai et al., 2023; Garrido et al., 2023; Gupta et al., 2023; Everett et al., 2024). (d) Most works have not validated the performance of different types and scales of backbone models within their frameworks, which is concerning as a general-purpose pre-training framework.

DDCL (Wang et al., 2024) differs from the aforementioned ECL frameworks that rely on equivariant-specific architectures and pretext task designs. It adopts a representation disentanglement approach, explicitly splitting the backbone's output into distortion-invariant and distortion-variant representations, and introduces an orthogonal loss function to adaptively disentangle the distortion-variant representation. By explicitly leveraging both distortion-variant and distortion-invariant semantic information, DDCL significantly improves training efficiency and robustness. More importantly, since DDCL adaptively extracts distortion-variant representations, it readily adapts to augmentation strategies of arbitrary complexity. However, our in-depth study of DDCL reveals several drawbacks, particularly concerning its use of orthogonal loss. Although utilizing

Table 1: Order of magnitude of the parameters and output representations of the projection head during pre-training in ImageNet-1K by the DDCL (w/ and w/o proposed $L_{PReg}$). $h_{V/I}$ and $z_{V/I}$ refer to the parameters of the head and representations respectively. All values are scaled by $\log_{10}(\cdot)$.

| Methods | DDCL | | | | DDCL w/ $L_{PReg}$ | | | |
|---|---|---|---|---|---|---|---|---|
| Epochs | $h_V$ | $h_I$ | $z_V$ | $z_I$ | $h_V$ | $h_I$ | $z_V$ | $z_I$ |
| 10 | -1.46 | 2.04 | -5.08 | -0.32 | 1.87 | 1.87 | -0.23 | -0.45 |
| 50 | -9.63 | 2.13 | -9.84 | -0.38 | 1.91 | 1.91 | -0.26 | -0.53 |
| 100 | -17.1 | 2.09 | -13.8 | -0.53 | 1.85 | 1.85 | -0.20 | -0.71 |
| 200 | -21.1 | 1.83 | -18.6 | -1.46 | 1.60 | 1.60 | -1.37 | -1.87 |
| Status | Collapse / Trivial Solution | | | | Similar projection | | | |

orthogonal loss to disentangle distortion-variant representations from different augmented views is reasonable, we find that it may lead to trivial solutions. Specifically, the projection head may collapse into a null space, especially when training on large-scale datasets, resulting in zero loss without achieving the intended mapping of orthogonal vectors in the latent space. This leads to an unstable training process, making DDCL difficult to employ effectively. Moreover, DDCL's performance across different backbone models has not been explored, leaving its generalizability unverified.

## 3 PROPOSED METHODS

Inspired by DDCL, we follow its methodology and employ an orthogonal loss function to supervise equivariant representations in the latent space for the ICL framework. In addition, based on the framework of DDCL, we propose a novel regularization loss for parameters of the projection head to address the instability of DDCL. Furthermore, we incorporate DINO (Caron et al., 2021) as the framework because it is not only a widely recognized method but also stable for more mainstream backbone models (Morningstar et al., 2024). By integrating equivariant representations into DINO through our stabilized DDCL, we propose a novel equivariant-based contrastive learning method, CLeVER. Moreover, we validate the training efficiency and robustness of CLeVER across various mainstream backbone models (ResNet, ViT and VMamba).

### 3.1 MAKE DDCL STABLE

DDCL (Wang et al., 2024) explicitly splits the output representation of the backbone model into distortion-invariant and distortion-variant representations. The contrastive and orthogonal losses are used to supervise the pairwise distortion-invariant and pairwise distortion-variant representations across different views during the contrastive process, respectively. The formula for this process is formulated as follows:

$$z_I^{(1,2)}, z_V^{(1,2)} = f(t_{1,2} \circ I) \tag{1}$$

$$L_I = L_{CL}(h_I(z_I^{(1)}), h_I(z_I^{(2)})) = -Similarity(h_I(z_I^{(1)}), h_I(z_I^{(2)})) \tag{2}$$

$$L_V = L_{Orth}(h_V(z_V^{(1)}), h_V(z_V^{(2)})) = h_V(z_V^{(1)}) \cdot h_V(z_V^{(2)}) \tag{3}$$

$$L_{DDCL} = \alpha L_I + \beta L_V \tag{4}$$

where $f(\cdot)$ is the backbone model of contrastive learning. The subscripts $I$ and $V$ refer to the variables or functions used for distortion-invariant and distortion-variant representations, respectively, and superscripts 1 and 2 represent two views of the same sample. $z$ and $h$ refer to the representation in the latent space and the projection head in the pretext task, respectively.

Analyzing the loss function of the distortion-variant representation of DDCL (*i.e.*, Eq. 3), we find that DDCL attempts to de-correlate the projected vectors of pairwise distortion-variant representations by making them orthogonal to each other. However, the orthogonality of $h_V(z_V^{(1)})$ and

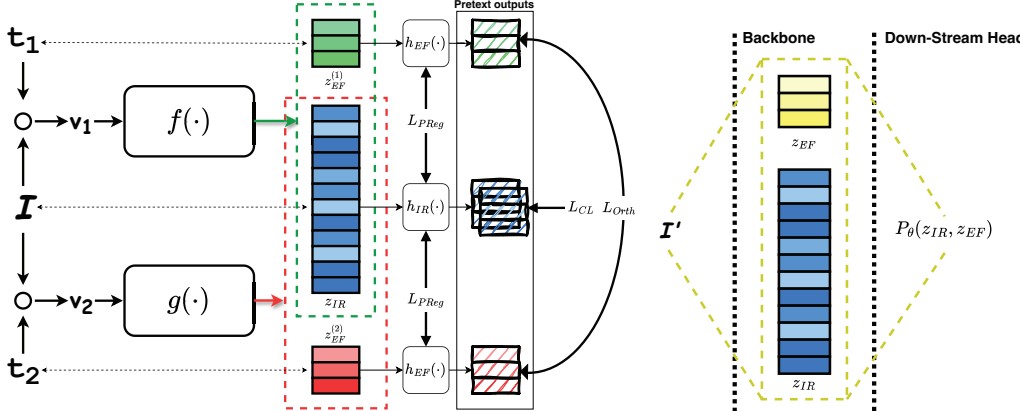

(a) Overview of the pre-training process for CLeVER.  (b) Inference in downstream tasks.

Figure 2: A brief overview of CLeVER. (a) $f(\cdot)$ and $g(\cdot)$ are backbone models. In DINO, they are EMA-based (Exponential Moving Average) teacher-student relationships. All $z_{EF}$ represent Equivariant Factors in the latent space corresponding to transformation operations $t_1$ and $t_2$, and $z_{IR}$ is denotes invariant representation of the invariant semantics in the latent space. $h$ represents the projection head used in the pretext task. In CLeVER, the loss of contrastive learning ($L_{CL}$) of the baseline method, the loss of orthogonality ($L_{Orth}$), and the projection regularization loss ($L_{PReg}$) are used. (b) In downstream tasks, the disentangled invariant representation and equivariant factor from the pre-trained backbone are incorporated for inference and prediction.

$h_V(z_V^{(2)})$ is not sufficiently necessary for $L_{Orth}$ to reach zero. We find that when training DDCL on large-scale datasets, the parameter values of the projection head ($h_V(\cdot)$) tend to zero, generating zero values for the projected vectors ($h_V(z_V^{(1)})$ and $h_V(z_V^{(2)})$). This trivial solution should be considered as a collapsed projection to a null space rather than achieving orthogonality between representations. Consequently, the split distortion-variant representations may not be effectively supervised and disentangled as expected.

To address the issue of trivial solutions, we introduce a novel regularization loss $L_{PReg}$ for parameters of the projection head. This loss function aligns the parameter magnitudes of $h_V$ and $h_I$, thereby preventing the collapse of $h_V$. The proposed loss function is formulated as follows:

$$L_{PReg} = L_1(\|h_V\|, \|h_I\|) = |\|h_V\| - \|h_I\||  \qquad (5)$$

where $L_1(\cdot)$ is the L1 loss function, $\|\cdot\|$ refers to the L2 norm. As demonstrated in Table 1, $L_{PReg}$ effectively stabilize DDCL by preventing training collapse and avoiding trivial solutions.

## 3.2 CLeVER

To introduce equivariant representations into contrastive learning and thereby improve the training efficiency, robustness, and generalizability of the backbone model, we revisit the definition of equivariance. Given a transformation group $T$ with group actions $t \triangleright_X$ and $t \triangleright_Y$ in domain $X$ and co-domain $Y$, respectively, we consider a function $f : X \to Y$ to be $T$-equivariance when it satisfies Eq. 6. We call it $T$-equivariance when $f$ satisfies Eq. 7. Obviously, $T$-invariance is a trivial case of $T$-equivariance (when $t \triangleright_Y := id_Y$).

$$f(t \triangleright_X x) = f(x) \quad \forall t \in T, x \in X  \qquad (6)$$

$$f(t \triangleright_X x) = t \triangleright_Y f(x) \quad \forall t \in T, x \in X  \qquad (7)$$

Assuming that the group $T$ has another group operation $t \triangleright'_Y = id_Y$ (identity operation) in the co-domain, Eq. 7 can be modified as follows:

$$f(t \rhd_X x) = t \rhd_Y f(x) = t \rhd_Y (t \rhd'_Y f(x)) \quad \forall t \in T, x \in X \tag{8}$$

This formulation indicates that when the transformation $t \in T$ perturbs the input $x \in X$ via the group action $t \rhd_X$, we can achieve $T$-equivariance by appropriately defining the group action $t \rhd_Y$ on $Y$. This allows us to attribute the effect of $t$ to the representation $f(x)$, which is $T$-invariant under the identity action $t \rhd'_Y$ in the co-domain $Y$ (latent space).

As illustrated in Fig. 2(a), we refer to the framework design of DDCL to explicitly split the representations extracted from the backbone model into **Invariant Representations** ($z_{IR}$) and **Equivariant Factors** ($z_{EF}$) according to a separation ratio. Furthermore, $z_{IR}$ and $z_{EF}$ are supervised, respectively, using the contrastive loss ($L_{CL}$, based on DINO) and orthogonal loss ($L_{Orth}$), with the help of the projection regularization loss ($L_{PReg}$). The group action $t \rhd_Y$ of the group $T$ in the co-domain $Y$ is realized as a concatenation operation, and a trainable neural network that is parallel to and shares some parameters with the backbone model $f$. We name the framework CLeVER, an abbreviation for Contrastive Learning Via Equivariant Representation. The formulas are given as follows:

$$(z_{IR}^{(1,2)}, z_{EF}^{(1,2)}) = t_{1,2} \rhd_Y f(x) = f(t_{1,2} \rhd_X x) \quad \forall t \in T, x \in X \tag{9}$$

$$L_{CL} = CE(Softmax(h_{IR}(z_{IR}^{(1)})), Softmax(h_{IR}(z_{IR}^{(2)}))) \tag{10}$$

$$L_{Orth} = Softmax(h_{EF}(z_{EF}^{(1)})) \cdot Softmax(h_{EF}(z_{EF}^{(2)})) \tag{11}$$

$$L_{PReg} = |\|h_{EF}\| - \|h_{IR}\|| \tag{12}$$

$$L_{Total} = \alpha L_{CL} + \beta L_{Orth} + \lambda L_{PReg} \tag{13}$$

During training, CLeVER retains the principle of extracting representations invariant to augmentation operations, as employed in ICL approaches. Moreover, it incrementally extracts representations that capture the effects of distortions or perturbations (Equivariant Factors) in a learnable manner. Thus, CLeVER provides information about perturbations without introducing inductive biases or prior assumptions (e.g., sensitivity or robustness to specific perturbations). CLeVER explicitly splits the extracted representations into Invariant Representations ($z_{IR}$) and Equivariant Factors ($z_{EF}$). Consequently, during inference (e.g., for classification tasks), the downstream prediction head performs a joint probabilistic prediction, i.e., $P_\theta(z_{IR}, z_{EF})$, based on $z_{IR}$ and $z_{EF}$, as illustrated in Fig. 2(b). This joint modeling allows downstream tasks to leverage both invariant and equivariant information, enhancing the model's robustness and generalization. Furthermore, we utilize Equivariant Factors to refer to the representations containing perturbation information ($z_{EF}$) since, unlike other CL methods designed to address only a single purpose, our experiments demonstrate that leveraging $z_{EF}$ achieves better performance on both augmentation invariance and sensitivity across experimental tasks (Tables 5 and 6 in Section 4.4).

### 3.3 MAKE ALL BACKBONES CLEVER

To comprehensively validate the generalizability of CLeVER, we select three representative backbone models: ResNet (He et al., 2016), ViT (Dosovitskiy et al., 2020), and VMamba (Liu et al., 2024), based on convolutional operators, self-attention mechanisms, and selective state space models, respectively. We also employ various sizes of backbone models, pre-training datasets, and downstream datasets to investigate CLeVER's training efficiency, performance, and robustness. We primarily utilize DINO (Caron et al., 2021) as the foundational framework due to its stability and support for more mainstream backbone models. Notably, CLeVER is fully adaptive and requires no augmentation-specific modifications based on DINO framework. This suggests that CLeVER can enrich the equivariance of backbone models by increasing the complexity of augmentation strategies and incorporating a wider variety of transformations.

Table 2: Comparison of linear performance of models pre-trained on IN-100. Approaches labeled *w/o R.* were pre-trained without rotated images and thus cannot handle rotations during inference. The green numbers represent performance increases over the corresponding baselines.

| Methods | Epoch | Handle R. | Backbones | #Params | GFLOPs | Top-1 | Top-5 |
|---|---|---|---|---|---|---|---|
| SimCLR *w/o R.* (Chen et al., 2020a) | 200 | ✗ | ResNet50 | 23.5M | 4.14G | 73.6 | - |
| SimCLR (Chen et al., 2020a) | 200 | ✓ | ResNet50 | 23.5M | 4.14G | 72.9 | - |
| Debiased *w/o R.* (Chuang et al., 2020) | 200 | ✗ | ResNet50 | 23.5M | 4.14G | 74.6 | 92.1 |
| BYOL *w/o R.* (Grill et al., 2020) | 200 | ✗ | ResNet50 | 23.5M | 4.14G | 76.2 | 93.7 |
| MoCo *w/o R.* (He et al., 2020) | 200 | ✗ | ResNet50 | 23.5M | 4.14G | 73.4 | - |
| MoCo v2 *w/o R.* (Chen et al., 2020b) | 200 | ✗ | ResNet50 | 23.5M | 4.14G | 78.0 | - |
| MoCo v2 (Chen et al., 2020b) | 200 | ✓ | ResNet50 | 23.5M | 4.14G | 72.0 | - |
| RefosNet (Bai et al., 2023) | 200 | ✓ | ResNet50 | 23.5M | 4.14G | 80.5 | 95.6 |
| | 200 | ✓ | ViT-Tiny | 5.5M | 1.26G | 66.2 | 89.0 |
| | 200 | ✓ | ResNet18 | 11.2M | 1.83G | 71.5 | 91.8 |
| DINO (Caron et al., 2021) | 200 | ✓ | ViT-Small | 21.7M | 4.61G | 73.2 | 92.7 |
| | 200 | ✓ | ResNet50 | 23.5M | 4.14G | 78.4 | 94.9 |
| | 200 | ✓ | VMamba-Tiny | 29.5M | 4.84G | 80.9 | 95.7 |
| | 200 | ✓ | ViT-Tiny | 5.5M | 1.26G | $68.7_{+2.5}$ | 90.7 |
| | 200 | ✓ | ResNet18 | 11.2M | 1.83G | $74.2_{+2.7}$ | 92.9 |
| **CLeVER (Ours)** | 200 | ✓ | ViT-Small | 21.7M | 4.61G | $75.7_{+2.5}$ | 93.6 |
| | 200 | ✓ | ResNet50 | 23.5M | 4.14G | $79.1_{+0.7}$ | 95.4 |
| | 200 | ✓ | **VMamba-Tiny** | 29.5M | 4.84G | $83.0_{+2.1}$ | **96.4** |
| Simsiam (Chen & He, 2021) | 500 | ✓ | ResNet50 | 23.5M | 4.14G | 79.7 | 94.9 |
| DDCL (Wang et al., 2024) | 500 | ✓ | ResNet50 | 23.5M | 4.14G | 80.0 | 95.0 |
| DDCL w/ $L_{PReg}$ (Ours) | 500 | ✓ | ResNet50 | 23.5M | 4.14G | $80.7_{+1.0}$ | 95.2 |
| | 500 | ✓ | ViT-Tiny | 5.5M | 1.26G | 70.2 | 91.4 |
| | 500 | ✓ | ResNet18 | 11.2M | 1.83G | 75.3 | 93.6 |
| DINO (Caron et al., 2021) | 500 | ✓ | ViT-Small | 21.7M | 4.61G | 76.0 | 93.9 |
| | 500 | ✓ | ResNet50 | 23.5M | 4.14G | 79.6 | 94.8 |
| | 500 | ✓ | VMamba-Tiny | 29.5M | 4.84G | 83.2 | 96.0 |
| CLeVER w/o $L_{PReg}$ | 500 | ✓ | ViT-Small | 21.7M | 4.61G | $76.3_{+0.3}$ | 93.4 |
| | 500 | ✓ | ViT-Tiny | 5.5M | 1.26G | $74.1_{+3.9}$ | 93.1 |
| | 500 | ✓ | ResNet18 | 11.2M | 1.83G | $78.1_{+2.8}$ | 94.3 |
| **CLeVER (Ours)** | 500 | ✓ | ViT-Small | 21.7M | 4.61G | $77.5_{+1.5}$ | 94.1 |
| | 500 | ✓ | ResNet50 | 23.5M | 4.14G | $80.0_{+0.4}$ | 95.2 |
| | 500 | ✓ | **VMamba-Tiny** | 29.5M | 4.84G | $83.9_{+0.7}$ | **96.5** |
| DINO (Caron et al., 2021) | 1000 | ✓ | ViT-Small | 21.7M | 4.61G | 76.3 | 93.3 |
| CLeVER (Ours) | 1000 | ✓ | ViT-Small | 21.7M | 4.61G | $78.3_{+2.0}$ | 94.6 |

## 4 EXPERIMENTS

### 4.1 EXPERIMENTAL SETTINGS

For pre-training, we utilize ImageNet-100 (IN-100) and ImageNet-1K (IN-1K). Due to computational constraints, IN-100 serves as our default dataset for training over 200 and 500 epochs (Sections 4.2 and 4.3). To further analyze the robustness of CLeVER (Section 4.3) and Equivariant Factors (Section 4.4), we report results using three data augmentation strategies: Basic Augmentation (BAug), Complex Augmentation (CAug), and High-Complexity Augmentation (CAug+). BAug refers to the data augmentation strategies used by DINO, which include color jittering, Gaussian blur, solarization, and multi-crop. CAug builds upon BAug by adding rotation, while CAug+ further extends CAug by incorporating elastic transformations (details are provided in Appendix A.1).

To comprehensively evaluate the generalization and practicality of CLeVER, we conduct downstream experiments (Section 4.5) on both in-domain and out-of-domain datasets (more implementation details are provided in Appendix A.2). To further assess the reliability of CLeVER, Appendix A.3 presents the performance gains achieved by pre-training on a large-scale dataset (IN-1k). Additionally, we perform ablation studies to confirm that the default hyperparameters used in CLeVER yield optimal performance (details are provided in Appendix A.4).

### 4.2 GENERALIZABILITY OF CLEVER

In Table 2, we use several mainstream backbone models to comprehensively examine the generalizability of CLeVER and the impact of equivariance across different backbones, comparing it with

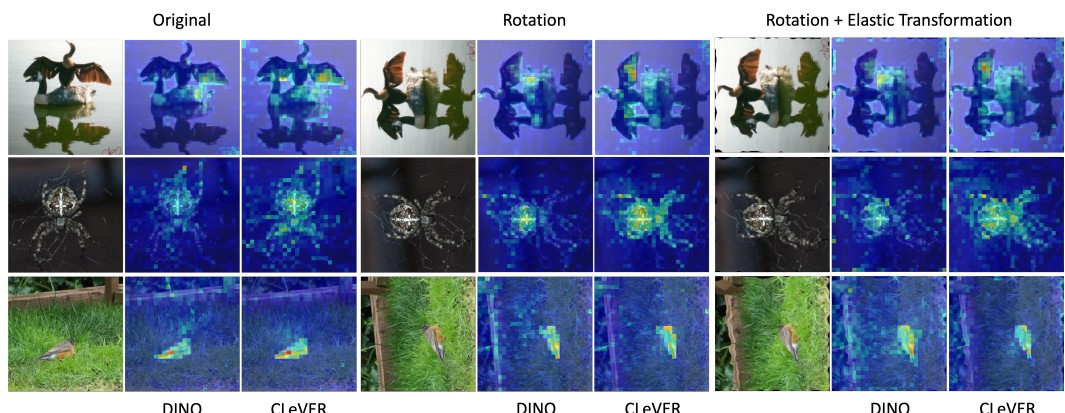

Original          Rotation          Rotation + Elastic Transformation

DINO   CLeVER       DINO   CLeVER       DINO   CLeVER

Figure 3: Visualization of self-attention under various augmentation settings.

Table 3: The effect of equivariance on the robustness of Simsiam.

| Methods | Orig. | CJ | CJ+Flip | CJ+Ro | CJ+Ro+ET |
|---|---|---|---|---|---|
| Trained by BAug | | | | | |
| Simsiam (Chen & He, 2021) | 81.9 | 81.3 | 81.4 | 50.3 | 27.3 |
| DDCL (Wang et al., 2024) | 82.2 | 81.6 | **81.6** | 50.0 | 26.8 |
| DDCL w/ $L_{PReg}$ (Ours) | **82.3** | **81.8** | **81.6** | 51.6 | 27.3 |
| Trained by CAug (w/ Ro) | | | | | |
| Simsiam | 79.7 | 79.0 | 79.0 | 77.0 | 51.9 |
| DDCL | 80.0 | 79.3 | 79.4 | 77.2 | 48.5 |
| DDCL w/ $L_{PReg}$ (Ours) | **80.7** | **80.2** | **80.0** | **77.6** | 48.1 |
| Trained by CAug+ (w/ Ro and ET) | | | | | |
| Simsiam | 78.6 | 77.7 | 77.7 | 75.1 | 74.1 |
| DDCL | 78.8 | 78.2 | 78.2 | 75.4 | 74.2 |
| DDCL w/ $L_{PReg}$ (Ours) | **79.8** | **79.0** | **79.3** | **77.0** | **75.5** |

Table 4: The effect of equivariance on the robustness of DINO.

| Methods | Orig. | CJ | CJ+Flip | CJ+Ro | CJ+Ro+ET |
|---|---|---|---|---|---|
| Trained by BAug | | | | | |
| DINO (Caron et al., 2021) | 78.2 | 77.6 | 77.2 | 52.7 | 41.0 |
| CLeVER w/o $L_{PReg}$ | **78.4** | 77.5 | 77.8 | 53.2 | 41.4 |
| CLeVER (Ours) | 78.3 | **77.8** | **78.1** | 53.4 | 41.2 |
| Trained by CAug (w/ Ro) | | | | | |
| DINO | 76.0 | 74.7 | 75.2 | 73.8 | 63.4 |
| CLeVER w/o $L_{PReg}$ | 76.3 | 75.7 | 75.4 | 74.6 | 64.5 |
| CLeVER (Ours) | **77.5** | **75.9** | **76.5** | **75.7** | 64.8 |
| Trained by CAug+ (w/ Ro and ET) | | | | | |
| DINO | 73.9 | 73.4 | 73.2 | 72.3 | 69.4 |
| CLeVER w/o $L_{PReg}$ | 74.3 | 73.6 | 74.0 | 73.1 | 70.7 |
| CLeVER (Ours) | **75.2** | **74.0** | **74.5** | **73.8** | **71.7** |

other state-of-the-art ICL and ECL approaches. The results show that our proposed $L_{PReg}$ enhances DDCL. Moreover, CLeVER improves the performance of the DINO framework across various types and scales of backbone models, achieving gains of 0.7–2.7% at 200 and 0.4–3.9% at 500 epochs. It is worth noting that smaller-scale models benefit more significantly from CLeVER. Comparing the performance between CLeVER and it *w/o* $L_{PReg}$ further emphasizes the importance of performance of projection and Equivariant Factor extraction. Furthermore, with the regularization loss, CLeVER exhibits continuous performance improvements as the training epochs increase from 200 to 1000. Notably, VMamba (Liu et al., 2024), a recently proposed backbone model, can be effectively integrated into our framework. Fig. 1(b) and Table 2 further demonstrate that VMamba achieves the best performance when introducing equivariance within the CLeVER framework. This suggests that the integration of Equivariant Factors shows superiority in maximizing the potential of innovative backbone architectures like VMamba.

## 4.3 ROBUSTNESS OF EQUIVARIANCE

To validate the positive impact of equivariance on the robustness of the backbone model, we use perturbed test data in the linear evaluation of the backbone model (pre-trained for 500 epoch with perturbations). In these experiments, in addition to CLeVER and DINO, we also include Simsiam (Chen & He, 2021), referring to DDCL, as a baseline to validate the effect of our proposed projection regularization. Orig. denotes no perturbation, CJ represents color jitter, Ro and ET denote rotation and elastic transformations, respectively.

In Table 3 and 4, the evaluation results suggest that our proposed projection regularization loss enhances the performance of robustness of ICL framework by preventing training collapse. With Simsiam as the baseline, introducing equivariance improves the performance of the backbone under the perturbation of rotation and elastic transformation by about 26.7% and 48.2%, respectively. Similarly, by incorporating equivariance, CLeVER improves the performance of vanilla DINO under perturbations of rotation and elastic transformation by about 21.1% and 30.7%, respectively. The improved performance, especially when training on complex perturbations (*i.e.,* CAug and CAug+),

Table 5: Experiments of rotational invariance.

Table 6: Rotational sensitivity.

| Methods | Linear | | | Fine-tune | | |
|---|---|---|---|---|---|---|
| | Orig. | Ro.(90°) | Ro.(180°) | Orig. | Ro.(90°) | Ro.(180°) |
| DINO | 65.3 | 62.6 | 61.3 | 76.9 | 68.5 | 65.6 |
| CLeVER w/o $L_{PReg}$ | 66.1 | 63.1 | 61.6 | 76.9 | 69.5 | 65.8 |
| CLeVER | **67.1** | **63.4** | **62.3** | **78.7** | **69.9** | **66.9** |

| Methods | Linear | Fine-tune |
|---|---|---|
| DINO | 52.2 | 75.2 |
| CLeVER w/o $L_{PReg}$ | 52.5 | 75.0 |
| CLeVER | **53.2** | **75.5** |

Table 7: Semantic analysis of Equivariant Factors (EF) and Invariant Representations (IR).

| Methods | Representations | Orig. | CJ | CJ+Flip | CJ+Ro |
|---|---|---|---|---|---|
| DINO | Total | 76.0 | 74.7 | 75.2 | 73.8 |
| | Total | 76.3 | 75.7 | 75.4 | 74.6 |
| CLeVER w/o $L_{PReg}$ | IR | 76.2 | 75.4 | 75.3 | 74.4 |
| | EF | 25.5 | 24.7 | 25.0 | 23.7 |
| | Total | **77.5** | **75.9** | **76.5** | **75.7** |
| CLeVER | IR | 77.3 | 75.8 | 76.4 | 75.2 |
| | EF | 4.1 | 4.1 | 4.0 | 3.7 |

indicates the promise of applying CLeVER to natural images and other practical scenarios involving more sophisticated information. Fig. 3 provides a visualization comparison of self-attention in ViT-Small. We observe that even when trained with the most complex augmentation setting (Rotation + Elastic Transformation), the proposed CLeVER learns more meaningful attention maps. The focused regions show high similarity across augmentations of different complexity, demonstrating the evidence of introducing perturbation-related information for outperformance. More detailed attention map comparisons corresponding to different attention heads are shown in Appendix A.5.

### 4.4 ANALYSIS OF EQUIVARIANT FACTORS

To explore the role of Equivariant Factors during inference, we employ the backbone pre-trained on CAug for 500 epochs and perform inference on OxfordPet (Parkhi et al., 2012), for both rotational invariance and rotational sensitivity testing. We use OxfordPet instead of IN100 because the faces and bodies of pets in this dataset are vertical, unlike the latter where the images themselves are tilted at different angles. In the rotational invariance test, the images are randomly rotated between $\pm 90°$ and $\pm 180°$, then used for evaluation in a downstream classification task to assess how well the model maintains classification performance despite the rotations. For the rotational sensitivity test, we perform 4-fold rotation predictions/classifications (90°, 180°, 270°, and 360°). We evaluate the accuracy of the model's predicted rotation angles as a downstream task. This assessment indicates the backbone's ability to recognize and be sensitive to rotational transformations.

The experiments in Tables 5 and 6 demonstrate that the Equivariant Factors extracted by CLeVER do not introduce inductive biases as vanilla architectures do. Instead, they provide perturbation-related information. Unlike most existing ECL frameworks designed for a single purpose, the equivariant information from CLeVER can be utilized in both ways depending on the requirements of downstream tasks. This flexibility results in both improved rotational invariance or rotational sensitivity.

Since the representations of Invariant Representation (IR) and Equivariant Factors (EF) are split and explicitly supervised by projection heads, we further separately use each representation in linear evaluation on IN-100 to explore the characteristics of each. In Table 7, we observe that the IR of CLeVER achieves slightly lower performance than the Total, indicating the effect of EF. Notably, when we use only EF for inference under different levels of perturbations, we find that the prediction accuracies of EF in CLeVER are around 3.7-4.1% (gray), compared to those without $L_{PReg}$ being around 23.7-25.5% (light gray). Lower EF accuracies and superior overall and IR inference performance provide evidence that the EF extracted by CLeVER, facilitated by $L_{PReg}$, contain less invariant semantic information and encapsulate more perturbation-related information as intended.

### 4.5 DOWNSTREAM TASKS

We use the pre-trained backbone models to evaluate the generalization ability and practicality of CLeVER on both in-domain and out-of-domain downstream tasks. The results in Table 8 demonstrate that CLeVER improves the efficiency of in-domain semi-supervised learning compared to DINO. In addition, in out-of-domain downstream classification tasks, CLeVER provides more significant improvements, especially when pre-trained with complex augmentation strategies.

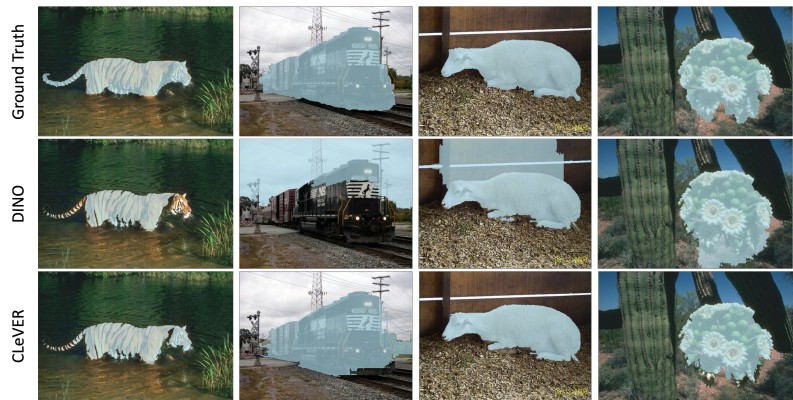

Figure 4: Qualitative performance of unsupervised saliency segmentation task.

Table 8: In-domain and out-of-domain downstream classification tasks.

| Methods | In-domain Semi. | | Out-of-domain Downstream | | | |
|---|---|---|---|---|---|---|
| | 1% | 10% | CUB200 | Flowers102 | Food101 | OxfordPet |
| | | | Trained by BAug | | | |
| DINO | 57.9 | **75.1** | **62.4** | **80.6** | 82.2 | 79.0 |
| CLeVER | **59.5** | 75.0 | 61.6 | 79.4 | **82.4** | **79.7** |
| | | | Trained by CAug (w/ Ro) | | | |
| DINO | 53.5 | 72.7 | 62.4 | 80.2 | 82.9 | 76.2 |
| CLeVER | **56.3** | **74.9** | **63.2** | **80.4** | **83.0** | **78.1** |

Table 9: Unsupervised downstream segmentation tasks.

| Methods | Video object seg. | | | Unsupervised saliency seg. | | | | | |
|---|---|---|---|---|---|---|---|---|---|
| | DAVIS 2017 | | | ECSSD | | DUTS | | DUT_OMRON | |
| | $(J\&F)_m$ | $J_m$ | $F_m$ | IoU | Acc. | IoU | Acc. | IoU | Acc. |
| | | | | Trained by BAug | | | | | |
| DINO | **0.594** | **0.582** | **0.606** | 0.657 | 0.866 | 0.431 | 0.789 | 0.427 | 0.780 |
| CLeVER | 0.592 | 0.577 | **0.606** | 0.673 | 0.877 | 0.440 | 0.801 | 0.443 | 0.787 |
| | | | | Trained by CAug (w/ Ro) | | | | | |
| DINO | 0.602 | 0.582 | 0.623 | 0.655 | 0.867 | 0.426 | 0.784 | 0.417 | 0.766 |
| CLeVER | **0.607** | **0.586** | **0.628** | **0.688** | **0.888** | **0.446** | **0.803** | **0.447** | **0.789** |

To validate the performance of pre-trained attention in downstream segmentation tasks, we conduct unsupervised video target segmentation tests referring to DINO. We also perform unsupervised saliency segmentation tests using TokenCut (Wang et al., 2022). The results in Table 9 indicate that CLeVER significantly improves unsupervised segmentation performance. Moreover, incorporating complex augmentation strategies and equivariance notably enhances the backbone model's segmentation capabilities. Fig. 4 qualitatively demonstrates that our proposed CLeVER generates superior attention-based saliency segmentation results compared to those of DINO.

## 5 CONCLUSIONS

**Summary.** This paper introduces a projection regularization loss to mitigate the risk of training collapse and trivial solutions in equivariant contrastive learning. By integrating equivariant representations into the invariant-based contrastive learning framework, we propose CLeVER, a novel equivariant contrastive learning method. CLeVER provides perturbation-related information without introducing additional inductive biases, significantly improving the training efficiency, generalization, and robustness of mainstream backbone models across various types and scales. **Limitations and Future Works.** Currently, we observe that the Equivariant Factors extracted by CLeVER contain less semantic information, and they assist the model in achieving better performance on both augmentation invariance and sensitivity experiments. Despite the promising performance of CLeVER, the perturbation-related information extracted by the stabilized orthogonal loss and stored in the Equivariant Factors is not yet fully understood. Therefore, in future research, we plan to investigate methods to extract Equivariant Factors in a more interpretable manner, aiming to gain deeper insights into their contribution to the model's performance.

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

# A APPENDIX

## A.1 AUGMENTATION SETTINGS

Compared to default augmentation setting used in DINO (*i.e.*, BAug), the CAug has an additional "transforms.RandomRotation(degrees=(-90, 90))" for all input images, and the CAug+ has additional "transforms.RandomRotation(degrees=(-90, 90))" and "transforms.RandomApply([transforms.ElasticTransform(alpha=100.0)], p=0.5)" for all input images.

## A.2 DETAILED EXPERIMENTAL SETUPS

For in-domain downstream tasks (1% and 10% semi-supervised learning), we use the same dataset as in pre-training (IN-100 or IN-1k). For out-of-domain downstream tasks, we use CUB200 (Wah et al., 2011), Flowers102 (Nilsback & Zisserman, 2008), Food101 (Bossard et al., 2014), and OxfordPet (Parkhi et al., 2012) for downstream classification tasks. Additionally, for out-of-domain segmentation downstream tasks, we use DAVIS 2017 (Shi et al., 2015), ECSSD (Wang et al., 2017), DUTS (Yang et al., 2013), and DUT_OMRON (Wang et al., 2022) as test sets.

All pre-training experiments are conducted on four NVIDIA A100 (80G) GPUs, with experimental setups identical to those of DINO (Caron et al., 2021) and DDCL (Wang et al., 2024). Referring to DINO Caron et al. (2021), when pretraining the model, we use SGD with base lr = 0.001, initial weight decay = 0.04, momentum = 0.9, and a cosine decay schedule on both IN-1k and IN-100 datasets. We conduct all experiments with a batch size of 128 per GPU on four NVIDIA A100 (80G) GPUs (or a batch size of 256 per GPU on 2 A100 GPUs), following the linear scaling rule Goyal et al. (2017). For linear evaluation, we use a SGD optimizer with 100 epochs, lr = 0.002, weight decay = 0, momentum = 0.9, and batch size per GPU = 128. On the linear evaluation experiments, only the linear layer is trained. In addition, identical to DINO, we use a warm-up strategy for a more stable training process with 10 warm-up epochs. For fine-tune-based downstream experiments (semi-supervised learning with 1% and 10% labels and downstream classification tasks on CUB200 Wah et al. (2011), Flowers102 Nilsback & Zisserman (2008), Food101 Bossard et al. (2014) and OxfordPet Parkhi et al. (2012)), we use a SGD optimizer with 200 epochs, lr of backbone and linear layer = 0.001, weight decay = 0.0001, momentum = 0.9, with a batch size of 256 per GPU. If the experiments are conducted with a batch size of 128 per GPU on four NVIDIA GPUs, the memory is less than 40G per GPU and the training time is around 3.5 hours per 100 epochs for ViT-small on IN100 datasets (The training time is also related to the type of hard drive.)

Table 10: Performance comparisons in ImageNet-1k.

| Methods | Orig. | CJ | CJ+Flip | CJ+Ro | CJ+Ro+ET |
|---------|-------|-----|---------|-------|----------|
| | | Trained by BAug | | | |
| DINO | **73.5** | 72.6 | **72.8** | 48.2 | 32.6 |
| CLeVER | **73.5** | **72.7** | 72.6 | 48.0 | 31.8 |
| | | Trained by CAug (w/ Ro) | | | |
| DINO | 71.8 | 70.9 | 70.9 | 70.0 | 55.6 |
| CLeVER | **72.0** | **71.2** | **71.2** | **70.1** | 55.3 |
| | | Trained by CAug+ (w/ Ro and ET) | | | |
| DINO | 70.2 | 69.3 | 69.3 | 68.3 | 66.4 |
| CLeVER | **70.7** | **69.9** | **69.8** | **69.0** | **66.8** |

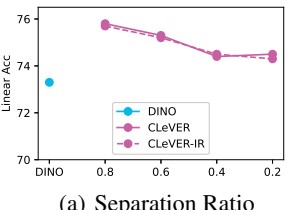 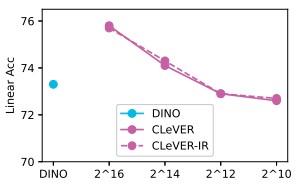 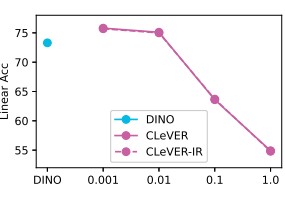

(a) Separation Ratio   (b) Output dimension of Head   (c) $\lambda$ coef. for regularization

Figure 5: Ablation studies on hyperparameters.

### A.3 CLeVER in Large-scale Dataset

We conduct robustness experiments with CLeVER on the large-scale dataset IN-1K to validate its reliability. In this experiment, we use ViT-Small as the backbone model, pre-training it for 100 epochs. Table 10 shows that on the large dataset, CLeVER still enhances the backbone model's robustness by increasing the complexity of the augmentation strategy, although the performance gain is less pronounced than on the medium-scale dataset IN-100. We attribute this to the substantial semantic information present in large datasets, which the backbone model can learn, making equivariance learning more challenging.

Furthermore, the results under the "Trained by CAug+" setting in Table 10 suggest that the gains from equivariance become increasingly significant as the perturbation complexity increases. This emphasizes the importance of incorporating complex augmentation strategies to maximize the robustness improvements offered by equivariance, even in large, information-rich datasets.

### A.4 Ablation Study

We perform ablation studies on some critical hyperparameters within CLeVER to ensure optimal configurations. Fig. 5(a) shows that the optimal separation ratio (*i.e.*, the ratio of the dimensions of $z_{IR}$ and $z_{EF}$) is 0.8. Fig. 5(b) demonstrates that the optimal choice of the output dimension for the projection head in CLeVER is the default $2^{16} = 65536$. Fig. 5(c) shows that the optimal weight $\lambda$ for $L_{PReg}$ is 0.001.

### A.5 Detailed Self-Attention Visualization

Figures 6, 7, and 8 provide detailed self-attention visualization maps for the six attention heads of ViT-Small, corresponding to Fig. 3. These figures illustrate that, compared to DINO, CLeVER learns more meaningful attention patterns across augmentation settings of varying complexity.

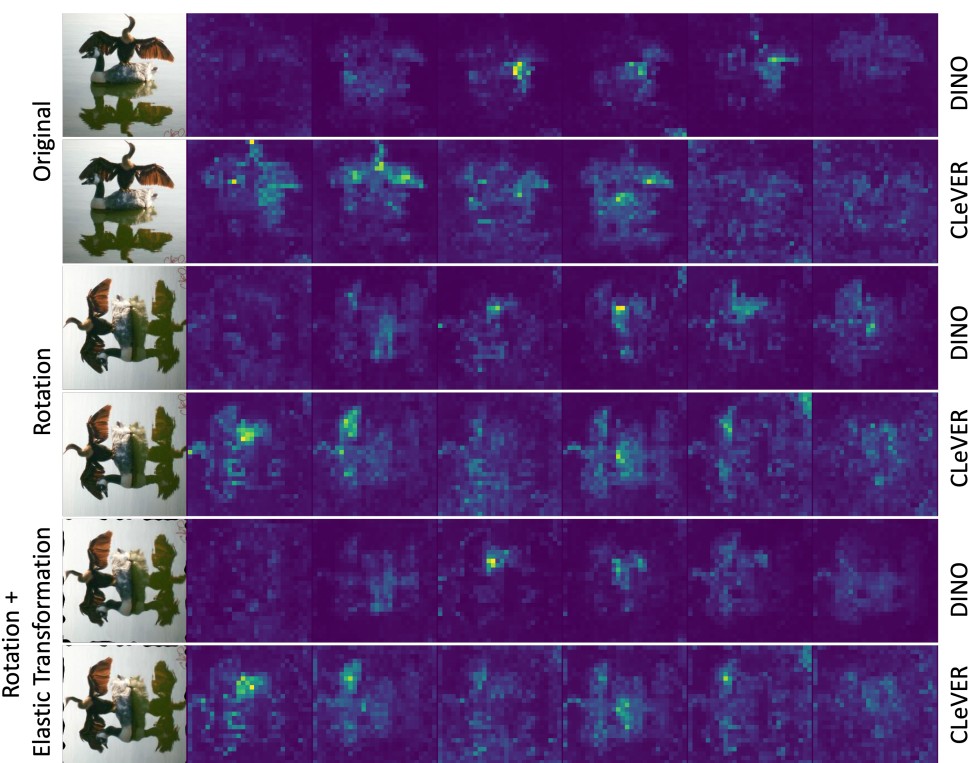

Figure 6: Detailed self-attention visualization maps from six attention heads (Example 1).

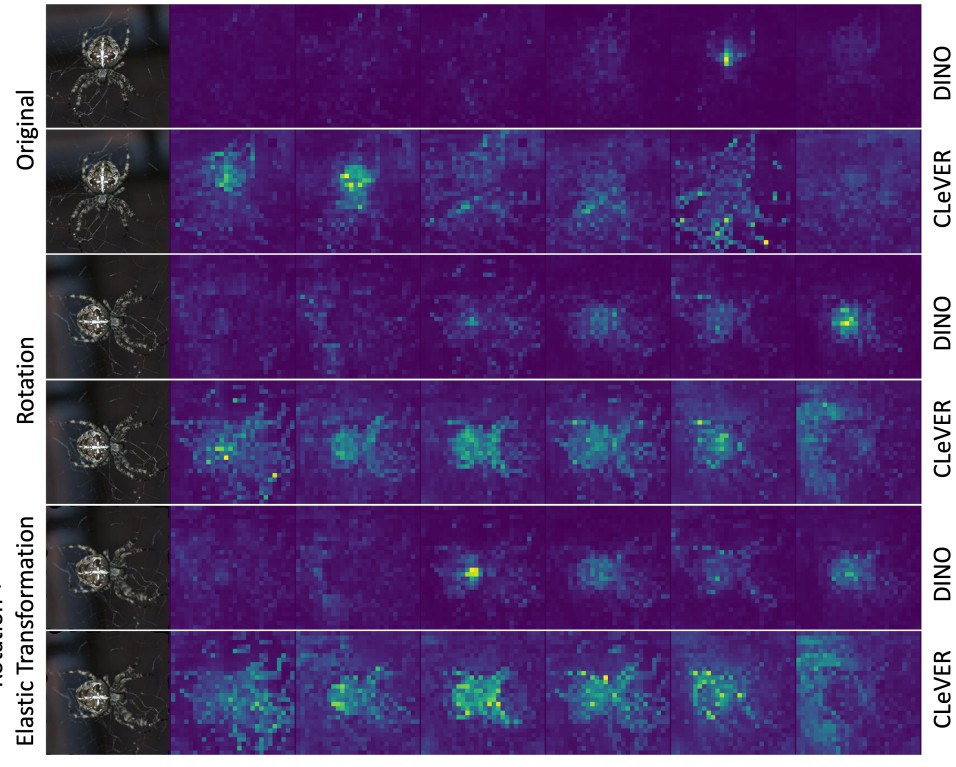

Figure 7: Detailed self-attention visualization maps from six attention heads (Example 2).

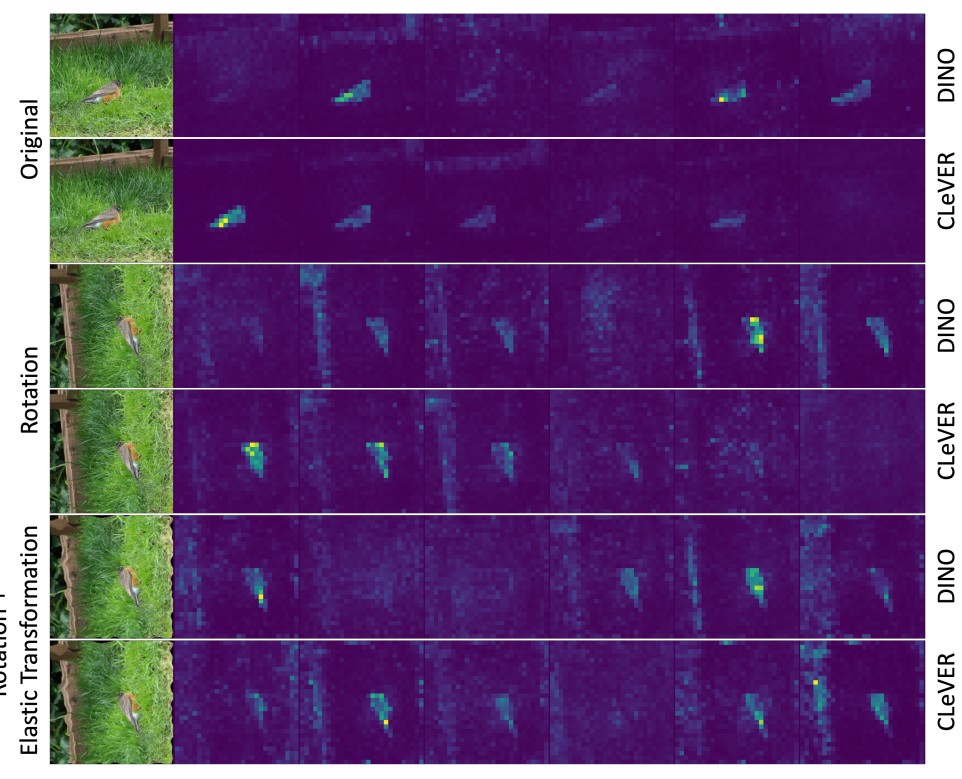

Figure 8: Detailed self-attention visualization maps from six attention heads (Example 3).

