# OpenReview forum: "Contrastive Learning Via Equivariant Representation"
_ICLR.cc/2025/Conference — Submitted to ICLR 2025_

### Official Review · Reviewer_VjBc · 2024-10-29

**Soundness:** 2
**Presentation:** 3
**Contribution:** 2
**Rating:** 5
**Confidence:** 3

**Summary:**

In this paper, the authors introduce a new regularization term into equivariant contrastive learning to avoid trivial solutions. Empirically, the regularization term improves the performance of contrastive learning across various downstream tasks and different backbones.

**Strengths:**

1. The paper is well-written and easy to follow. The main motivation, i.e., the original equivariant loss will lead to trivial solutions, is clear and the solution is straightforward.
2. The experiments are comprehensive, and the proposed objective shows benefits in different scenarios.

**Weaknesses:**

1. It seems that the main difference between CleVER and DDCL lies in the regularization term. However, is there any evidence to show that DDCL obtains trivial solutions in practice and obtains inferior performance? It would be better to add more discussions about the effectiveness of the regularization term and why it works.
2. In downstream tasks, the authors use a combination of equivariant and invariant factors. However, as we have no access to the properties of downstream tasks, how can we decide which kind of factions to rely on?
3. It seems that the empirical improvements are a little marginal, especially compared with DDCL. It would be better to add more discussions about the advantages of CleVER.
4. I think the ablation study on the balance of three terms in the pretraining objective is necessary. It would be better to show the advantages and disadvantages of invariant and equivariant features.

**Questions:**

See Weaknesses.

---

### Official Review · Reviewer_hk6G · 2024-10-31

**Soundness:** 3
**Presentation:** 1
**Contribution:** 2
**Rating:** 3
**Confidence:** 4

**Summary:**

This paper introduces an Equivariant contrastive learning method called CLeVER. Inspired by DDCL, CLeVER disentangles representations into Invariant Representations and Equivariant Factors. It performs contrastive learning on the Invariant Representations while ensuring orthogonality of the Equivariant Factors. A regularization loss is utilized to prevent trivial solutions. Experimental results demonstrate that this method achieves excellent performance.

**Strengths:**

- This paper is well-motivated. It is important to explore equivariance of contrastive learning.
- CLeVER proposed in this paper is concise and intuitive.
- Extensive experiments demonstrate the effectiveness of CLeVER.

**Weaknesses:**

1. The innovation of CLeVER is minimal. CLeVER without regularization loss is very similar to DDCL. From my perspective, CLeVER without regularization loss simply changes the way $L_{CL}$ and $L_{Orth}$ are computed in DDCL. Moreover, the computation of $L_{CL}$ and $L_{Orth}$ is also very common.
2.  Section 3.3 primarily introduces the backbone used by CLeVER. I believe this section is irrelevant to the method and should be moved to Section 4 as part of the experimental settings.
3. Table 2 conveys too much information, making it difficult to grasp the intended message in Section 4.2.
    - I suggest breaking down Table 2 into several smaller tables based on the discussions in Section 4.2. Each smaller table should focus on a specific aspect of the research, such as the study of the backbone, a comparison of DDCL (CLeVER) with or without $L_{PReg}$, a comparison between DDCL and CLeVER, a comparison of CLeVER training epochs, and so on.
    - Additionally, there is some redundant information in Table 2 that is not discussed in the main text, such as the results of SimCLR, Debiased, BYOL, MoCo, MoCo V2, and RefosNet.
4. Comparing only CLeVER and DDCL in this paper is insufficient. I believe it would be better to compare CLeVER with more ECL methods, such as [1], [2], [3].

    [1] Dangovski, Rumen, et al. "Equivariant contrastive learning." *arXiv preprint arXiv:2111.00899* (2021).

    [2] Xie, Yuyang, et al. "What should be equivariant in self-supervised learning." *Proceedings of the IEEE/CVF Conference on Computer Vision and Pattern Recognition*. 2022.

    [3] Gupta, Sharut, et al. "Structuring representation geometry with rotationally equivariant contrastive learning." *arXiv preprint arXiv:2306.13924* (2023).

**Questions:**

1. Section 3.2 mentions "we consider a function f: X → Y to be T-equivariance when it satisfies Eq. 6. We call it T-equivariance when f satisfies Eq. 7." I believe that satisfying Eq. 6 does not constitute T-equivariance. Is there a typo present here?
2. Table 3 and Table 4 respectively analyze the effect of equivariance on the robustness of SimSiam and DINO. However, why is DDCL used to analyze SimSiam while CLeVER is used to analyze DINO? I believe it would be more reasonable to analyze SimSiam and DINO using the same method.
3. In Table 3 and Table 4, some results are shaded in gray. What is the purpose of this shading? Additionally, the best results in the shaded columns are not bolded.

---

### Official Review · Reviewer_UQk5 · 2024-11-03

**Soundness:** 2
**Presentation:** 2
**Contribution:** 1
**Rating:** 3
**Confidence:** 5

**Summary:**

This study presents a novel equivariant contrastive learning framework, termed CLeVER, which is compatible with augmentation strategies of varying complexities across various mainstream contrastive learning backbone models. Specifically, CLeVER enhances the DDCL method by mitigating the instability in the training process that can result in trivial solutions.

**Strengths:**

This paper is motivated by the important problems with existing equivariant contrastive learning and introduces a regularization term designed to mitigate collapse problem in DDCL.

**Weaknesses:**

The novelty of the proposed method is limited, as the primary difference from DDCL lies solely in the incorporation of a regularization term, which is applicable only to DDCL. Considering this aspect, the contribution of the study appears significantly constrained.

Furthermore, there has been no analysis of the collapse phenomenon that occurs within the DDCL framework. The study does not provide a thorough examination of why the proposed regularization term is the most effective solution for addressing the collapse issue in DDCL.

The prevention of representation collapse in contrastive learning has been a longstanding area of investigation. However, there is currently a lack of comprehensive reviews summarizing these studies. It is essential to explicitly delineate how CLeVER distinguishes itself from existing research. Therefore, a detailed section on related works should be included to address these aspects.

Additionally, while the baseline framework employed is DINO, the effectiveness of the method has not been validated against alternative frameworks such as SimCLR or Barlow Twins.

Moreover, there is a lack of benchmark comparisons with other equivariant contrastive learning methods.

**Questions:**

How does addressing the collapse problem in DDCL contribute to the improvement of equivariance?

---

### Official Review · Reviewer_dxxT · 2024-11-03

**Soundness:** 3
**Presentation:** 3
**Contribution:** 2
**Rating:** 5
**Confidence:** 4

**Summary:**

This paper proposes CLeVER, an ECL framework which adds regularization to the decoupled contrastive learning representations that is previously proposed, encouraging similarity on the invariant part of positive pairs and orthogonality on their equivariant part, and regularizing on the distance between the normed logits of the two parts.

**Strengths:**

1. The proposed regularization does improve DINO across multiple backbones.

2. The analytical results and visualizations are helpful. The experiment section is thoughtful and covers various aspects for better evaluation.

3. This paper is easy to follow.

**Weaknesses:**

1. This paper proposes a novel regularization based on intuition, but does not discuss any theoretical insights. Considering Table 1, when minimizing $|\|\|h_V\|\|-\|\|h_I\|\||$, why $h_V$ is properly regularized, with values increasing toward that of $h_I$, but $h_I$'s do not decrease much, as moving toward $h_V$ also helps minimize the regularization loss? Moreover, the authors do not report such table on CLeVER for us to compare with Table 1 to check the improvements.

2. Based on Table 2 results, despite the continual improvements on DINO, the authors do not study the effectiveness and the ability to generalize of CLeVER on other contrastive methods besides DINO and DDCL. I am skeptical about whether it can work well with other similar methods such as Barlow Twins, SimSiam. Based on Sec 3, does CLeVER rely heavily on DDCL? I am afraid that the contribution of the proposed regularization is limited to improving DDCL and a few similar work. Moreover, 0.7 improvement on DDCL for one setting is not convincing, it would be more comprehensive if we could see how CLevER improves DDCL across various backbones and scales like for DINO. After all, DINO has been proposed for more than three years, and I also encourage the authors to consider more recent methods.

3. Thanks for the attention visualizations, but I am curious about any quantitative results and analysis. From the visualizations, I notice that CLeVER tends to have larger values and wider coverage on the object than DINO, which is great, but it also spills some attention to unrelated pixels, which is more obvious in supplementary visualizations for independent attention heads.

**Questions:**

Despite the demonstrated effectiveness of CLeVER regularization, I am majorly concern about its application to general contrastive learning methods. If it cannot be adopted by general methods and is dependent on certain methods and objectives, I am afraid that its contribution to the CL literature is limited. Moreover, the authors do not explain in depth the motivation for such regularization as I still have concerns. Please see the weakness section, and in short, I am majorly concerned about (2), then (1), and the authors may resolve (3) if time permits.

---

### Meta-Review · Area_Chair_mp7b · 2024-12-17

**Metareview:**

This paper proposes an equivariant-based contrastive learning framework to address limitations in invariant contrastive learning (ICL). The authors claim that the absence of augmentation-related information in ICL leads to inefficiencies and reduced robustness. CLeVER introduces a regularization term designed to decouple invariant and equivariant representations while maintaining orthogonality and minimizing the distance between their normed logits. Experiments demonstrate performance gains on baseline models such as DINO across downstream tasks and various backbones.

Reviewers point out that CLeVER is closely related to DDCL, with the primary difference being the addition of a regularization term. The theoretical motivation behind this term is not fully explained, leaving the contribution incremental. The authors are encouraged to add more explanations and thorough studies and comparisons to better elaborate the contributions.

**Additional Comments On Reviewer Discussion:**

Several common points were arised among the reviewers:

- Novelty and Contribution: Reviewers dxxT, UQk5, and hk6G highlighted that CLeVER's main innovation—introducing a regularization term—is incremental and primarily tied to DDCL. The authors did not provide sufficient explanations or evidence on why this regularization is theoretically motivated or effective.
- Generalizability: Reviewers questioned whether CLeVER can work with other methods like SimCLR or SimSiam. The authors did not present additional results or clarifications during the rebuttal phase.
- Experimental Scope: While CLeVER showed consistent improvements over DDCL, the margins were modest. Reviewers requested more benchmarks against other equivariant contrastive learning methods (e.g., works by Dangovski et al., Xie et al., Gupta et al.), which were missing.
- Attention Visualizations: Reviewer dxxT appreciated the visualizations but raised concerns about quantitative results and unintended "spilling" of attention to unrelated areas.

The authors did not provide a rebuttal to address these concerns and no further discussion is raised. Thus, I assume that these concerns do remain.

---

### Decision · Program_Chairs · 2025-01-22

Reject